# Dendronized fluorosurfactant for highly stable water-in-fluorinated oil emulsions with minimal inter-droplet transfer of small molecules

Mohammad Suman Chowdhury [1], Wenshan Zheng[2], Shalini Kumari[1], John Heyman [3], Xingcai Zhang [3], Pradip Dey [1], David A. Weitz[3]* & Rainer Haag [1]*

Fluorosurfactant-stabilized microfluidic droplets are widely used as pico- to nanoliter volume reactors in chemistry and biology. However, current surfactants cannot completely prevent inter-droplet transfer of small organic molecules encapsulated or produced inside the droplets. In addition, the microdroplets typically coalesce at temperatures higher than 80 °C. Therefore, the use of droplet-based platforms for ultrahigh-throughput combination drug screening and polymerase chain reaction (PCR)-based rare mutation detection has been limited. Here, we provide insights into designing surfactants that form robust microdroplets with improved stability and resistance to inter-droplet transfer. We used a panel of dendritic oligo-glycerol-based surfactants to demonstrate that a high degree of inter- and intramolecular hydrogen bonding, as well as the dendritic architecture, contribute to high droplet stability in PCR thermal cycling and minimize inter-droplet transfer of the water-soluble fluorescent dye sodium fluorescein salt and the drug doxycycline.

[1] Institut für Chemie und Biochemie, Freie Universität Berlin, Takustrasse 3, 14195 Berlin, Germany. [2] Department of Chemistry and Chemical Biology, Harvard University, Cambridge, MA 02138, USA. [3] School of Engineering and Applied Sciences, Department of Physics, Harvard University, 29 Oxford Street, Cambridge, MA 02138, USA. *email: weitz@seas.harvard.edu; haag@chemie.fu-berlin.de

Fluorosurfactant-stabilized, water-in-fluorinated-oil (w/o) droplets, with volumes of pico- to nanoliters, have facilitated a variety of powerful research techniques. Emulsions made with fluorosurfactant-containing fluorinated oil have been shown to be biologically inert and, due to fluorinated oil's capacity for dissolved oxygen, are suitable for cell culture. Accordingly, these emulsions have been exploited for numerous biological applications, including rapid parallel transcriptome profiling of thousands of cells with single-cell resolution[1,2], high-throughput drug screening[3–5], analysis of products secreted by individual cells[6], directed evolution of desired enzymes[7,8], and construction of synthetic cells[9]. Additionally, emulsion droplets are easily generated with highly monodisperse and reproducible droplet size, making them suitable to study different aspects of physics and chemistry[10–14]. These tiny reaction chambers are created, analyzed, and sorted at kHz rates, typically using polydimethylsiloxane (PDMS) microfluidic devices[4]. However, small organic molecules can exchange between adjacent droplets[4,12], limiting their utility for high-throughput screening applications. In addition, droplets are often unstable during thermal cycling, reducing reliability of the polymerase chain reaction, (PCR)[4,15], which is an extremely efficient method for nucleic acid amplification and is a critical step in most genetic analyses. The poor droplet integrity is often caused by the nature and type of the fluorosurfactant used to stabilize the droplet.

The non-ionic tri-block copolymer fluorosurfactant, PEG-PFPE$_2$ (EA surfactant from RAN Biotechnologies), made of poly (ethylene glycol) and perfluoropolyethers (PFPE), is widely used to stabilize the emulsion droplets within fluorinated oil[1,6,11,16,17]. However, during thermal cycling (PCR), a significant number of droplets merge. In addition, small molecules (~200 to ~500 Dalton) can easily pass between droplets[12,15]. The poor performance of PEG-PFPE$_2$ surfactant can partially be attributed to its synthesis limitations and to its structure. PEG molecules are also somewhat hydrophobic[18] and thermo-responsive[19–21], which can influence biological assays[18,20] and destabilize droplets that are subjected to temperature change, e.g., during PCR. In addition, due to the polydisperse nature of both PEG and PFPE, the exact molar amount of each molecule cannot be determined and the ratio of the molecules (1:2, respectively) during tri-block copolymer synthesis is often incorrect. Consequently, a mixture of di-block, tri-block, unmodified precursors, and ionically coupled surfactant molecules is created[11], generating batch-to-batch variation in the final product. Finally, surfactants with a tri-block structure can efficiently form micelles and bilayer vesicles, which can destabilize and act as carriers between drops.

Di-block surfactants do not suffer from these problems. Their synthesis can employ an excess of polar head group to ensure complete covalent coupling to the fluorinated tail. The unconjugated head group can then be removed by simple work-up, resulting in pure di-block surfactant, with no undesired products that may destabilize droplets. Structurally, because di-block surfactants cannot form bilayer vesicles, vesicle-mediated inter-droplet cargo transfer should not occur.

Although di-block fluorosurfactants have been synthesized with a variety of head groups, including carbohydrate-derivatives[22] and PEG of various lengths[23], these surfactants were inferior to the commonly used PEG600-PFPE$_2$ tri-block copolymer fluorosurfactant[11], justifying testing of alternative head groups. We hypothesize that use of dendritic head groups carrying hydroxy moieties, which readily form inter- and intra-molecular hydrogen bonds[24], might lead to high-performance surfactants. However, we rule out linear head groups based on our previous studies showing that a linear polyglycerol-based tri-block copolymer fluorosurfactant, despite having thirteen hydroxy groups, was unable to stabilize droplets[15]. Instead,

different architectures of head groups are needed to design new and effective fluorosurfactants.

Here, we describe a systematic design and testing of dendronized fluorosurfactants containing mono- or tri-glycerol polar head groups and fluoro-tails of three different lengths. Our two best-performing surfactants combined a four hydroxy-group-containing polar head group with fluorinated tails of low- or medium length (2 or 4 kDa). For microdroplet PCR and small molecule droplet-retention, these surfactants were superior to surfactants made with a polar head group lacking hydroxy groups and to the PEG-PFPE$_2$ surfactant.

## Results and discussion

**Synthesis of dendronized fluorosurfactants**. For the synthesis of the dendronized fluorosurfactants we use a facile amide coupling to covalently couple the dendritic tri-glycerol (dTG) moiety to the PFPE chains. By reacting the amine-functionalized oligo-glycerols with the activated carboxy terminus of the non-polar fluoro-chains, followed by deprotection of the acetal groups the fluorosurfactants were obtained in high yields of 75–85% (Fig. 1). It is worth mentioning that the unreacted protected oligo-glycerol derivatives are purely soluble in organic solvents such as dichloromethane (DCM), methanol, and tetrahydrofuran (THF). In addition, these organic solvents are immiscible with fluorinated solvents, including HFE7100 and HFE7500, and, due to their lower density, phase separate into a top layer. Thus, excess unreacted protected oligo-glycerol moieties can be easily removed by washing with excess of DCM. Deprotected oligo-glycerol derivates are soluble in THF or methanol, and additional washing with these solvents results in pure di-block fluorosurfactant, with no undesired products. We synthesized three surfactants that are consisted of tri-glycerol-based dendritic polar heads (dTG), which have four hydroxy (–OH) groups, and PFPE polymer chains with three different fluorine chain lengths of high (H), medium (M), and low (L) molecular weights (MW), providing H-dTG, M-dTG, and L-dTG, respectively, are shown in Fig. 1 (bottom panel). We also synthesized two surfactants that carry only two –OH groups in their non-ionic heads using glycerol-amine. The combination of mono-glycerol (G) with medium and low MW PFPE chains created M–G and L–G surfactants, respectively, as illustrated in Fig. 1 (top panel). These five di-block dendronized surfactants allow us to systematically investigate the effect of polar head group geometries and the number of –OH groups on droplet stability and performance under demanding conditions. The successful preparation of the fluorinated surfactants was confirmed by the amide (-NH-CO-) stretching peak from 1710 to 1740 cm$^{-1}$ in FT-IR spectroscopy. In contrast, an unmodified PFPE-COOH (Krytox) showed a strong IR band at 1775 cm$^{-1}$ (Supplementary Fig. 1). We used deionized water and a 2% (w/w) surfactant in HFE7500 oil to prepare w/o emulsions by bulk-mixing. Surprisingly, all five surfactants produced stable emulsions irrespective of the length of PFPE chains and the number of glycerol units. Furthermore, long-term incubation of the emulsions, prepared by tri-glycerol based surfactants, for about 30 days at room temperature in small glass vials did not cause any noticeable merging of the droplets. In contrast, mono-glycerol based surfactants stabilized the droplets for about a week.

**Emulsion droplet stability during PCR**. We investigated the ability of all five di-block surfactants to stabilize the microfluidically generated large droplets, ~95 micrometer (μm) diameter (~500 pL) and test their stability during PCR reactions between 60 °C and 98 °C. We chose relatively large droplets because droplet stability decreases with droplet size and we

**Fig. 1** Synthesis of fluorosurfactants with dendritic glycerol head groups. PFPE tails of two different fluoro-chain lengths (M, L) were conjugated to the mono-glycerol (G) polar head (upper drawing, light blue) to create surfactants M-G and L-G; PFPE tails of three different molecular weights (H, M, L) were conjugated to dendritic tri-glycerol (dTG) polar head to create surfactants H-dTG, M-dTG, and L-dTG (lower drawing, blue). These surfactants orient at the interface (deep gray) of water (light red) and fluorinated oil (gray) to stabilize picolitre-volume (pL) water-in-oil (W/O) emulsion droplets, as depicted in the picture

wanted a stringent test case, and we used the PEG-PFPE$_2$ surfactant as a reference standard. Moreover, complex droplet manipulations, for example in-droplet barcoding[1,25] and pico-injection[26,27] often use large droplets. Thus, using each of these six surfactants, at equal w/w concentrations (2%), we encapsulated PCR mix into monodisperse droplets of ~95 μm diameter that were stable at room temperature (Supplementary Fig. 2). After 35 cycles of PCR reaction between 60 °C and 98 °C, we found that all three surfactants containing the tri-glycerol dendron (dTG) as the polar head group effectively stabilized the drops during PCR reaction, irrespective of the PFPE chain lengths (Fig. 2b and Supplementary Fig. 3). Within this group, the L-dTG surfactant was superior to the H-dTG and M-dTG surfactants (Supplementary Fig. 4). We created droplets using these surfactants at roughly equimolar amounts, corresponding to 2% w/w H-dTG and 0.7% w/w L-dTG. Under these conditions, H-dTG droplets were more stable than L-dTG-stabilized droplets and the L-dTG-stabilized droplets coalesced upon collection and/or minimal external force applied to the droplet collection tubing, suggesting that the H-dTG longer fluorinated tails can provide more stability than shorter tails. Of note, droplets made with 2% w/w L-dTG were more stable than those formed with the PEG-PFPE$_2$ surfactant (Fig. 2d, Supplementary Figs. 3–4). However, droplets generated with 4% w/w PEG-PFPE$_2$ had superior post-PCR thermal stability than droplets stabilized with 2% w/w PEG-PFPE$_2$ (Supplementary Figs. 3–4). A quantitative analysis of the post-PCR droplet size distributions shows that dTG and PEG based reference surfactants ability to stabilize droplets follows this order: 2% L-dTG ≥2% M-dTG > 4% PEG-PFPE$_2$ ≥2% H-dTG > 2% PEG-PFPE$_2$ (Supplementary Figs. 3–4).

Droplets formed with the mono-glycerol (G)-based surfactants were also stable at room temperature. However, all the droplets merged during PCR, as illustrated in Fig. 2b, c and Supplementary Fig. 5b. Because the only structural difference between dTG and G polar groups is the number of hydroxy groups, 4 vs 2, respectively, this result clearly demonstrates the importance of inter- and intra-molecular hydrogen bonding. Furthermore, the head-group architecture also influences surfactant performance. Fluorosurfactants created with glucose derivatives, which also carry four hydroxy groups, whose primary difference from dTG-surfactants is a cyclic vs. a dendritic geometry, are unable to make stable emulsions[22]. We hypothesize that, mechanistically, the dTG's four hydrogen bond donors, and its dendritic architecture

collectively favors inter- and intramolecular H-bonding and provides greater emulsion stability.

We further characterized the performance of dTG-based surfactants by performing in-droplet click-chemistry to crosslink together the high-viscosity polymers dendritic poly(glycerol-sulfate) azide and homo-bifunctional PEG-cyclooctyne, generating spherical hydrogels. We prepared highly monodisperse droplets, carrying polymer precursors, at kHz rates with all three types of surfactant using a concentration of 2% (w/w) in HFE7500 oil (Supplementary Fig. 6). We incubated the emulsions to allow crosslinking, and then transferred the gels to aqueous solution. For all three surfactants, the isolated gels were monodisperse and no inter-microgel cross-linking was seen, proving that droplets did not merge during crosslinking and that the crosslinking was complete. This suggests that dTG-based surfactants, which have equally high performance when synthesized with PFPE tails of ~2, ~4, or ~6 kDa, tolerate a larger range of PFPE tail length than PEG-based fluorosurfactants, which function only if the PFPE chain is ~6–7.5 kDa[11,12,15,23].

These data show that, for a given fluorinated tail group, dTG containing surfactants can stabilize droplets better than the corresponding G containing surfactants. We attribute this to dTG's symmetrical architecture, its four H-bond donors and acceptors, and the flexibility provided by its two chiral centers. Tri-glycerol based dendronized surfactants' robust droplet-stabilizing activity prompted us to study its capacity to prevent small molecule inter-drop diffusion. This is of great interest, as it is extremely challenging to prevent leaching of small molecules to the neighboring microscopic droplets[4]. In addition, a surfactant that prevents substrate leakage will allow compartmentalization of small-molecule drugs, enabling in vitro high-throughput screening in very small volume droplets. Performing biological assays in picolitre-volume microscopic droplets instead of robot-assisted screening in microtiter plates would lead to dramatic cost savings due to low reagent consumption and, in cell-based assays, a significant reduction in number of cells required[7].

**Minimized inter-droplet transfer**. We studied the inter-droplet leakage of a small water-soluble dye, sodium fluorescein salt ($M_w = 376$ g mol$^{-1}$) by mixing populations of empty and dye-containing droplets and measuring the transfer of dye to the empty droplets. For each surfactant tested, we used a parallel drop

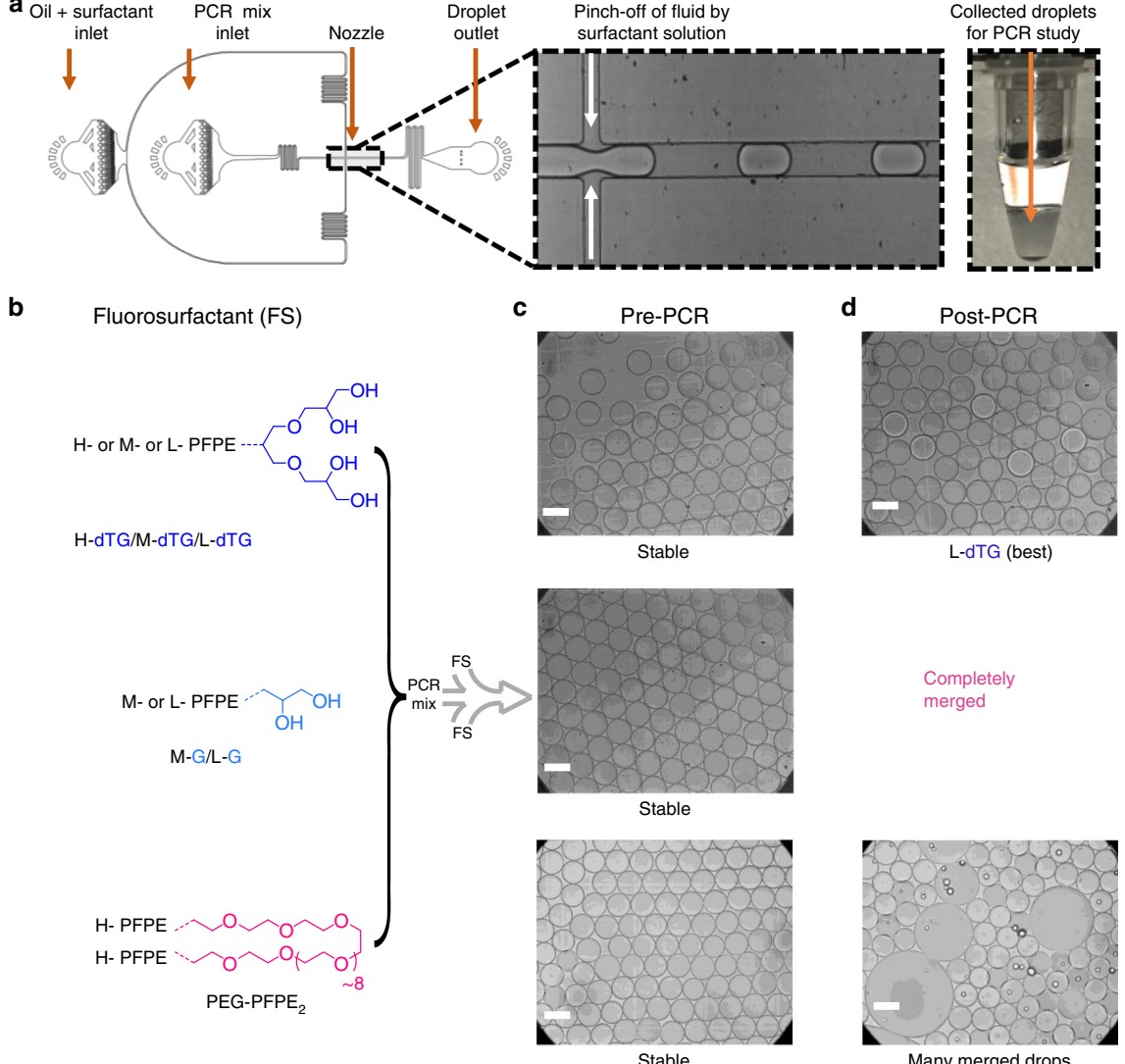

**Fig. 2 Polar head group geometry dictates droplet stability. a** Polydimethylsiloxane (PDMS) based microfluidic devices are used to prepare microdroplets containing PCR reagents. **b** The three tri-glycerol (dTG)-based surfactants (H-dTG, M-dTG, L-dTG), the two mono-glycerol (G)-based surfactants (M-G, L-G), and the PEG600-based fluorosurfactant, PEG-PFPE₂, were used to create droplets for stability testing (chemical structures in **b**). **c** Droplets produced using the tri-glycerol, the mono-glycerol, and the PEG600-based surfactants showed no merging during >24 h incubation at 4 °C (**c** showing pre-PCR droplets). **d** After 35 cycles of PCR, L-dTG-stabilized droplets showed almost no merging. In contrast, droplets generated using mono-glycerol (G)-based surfactants (M-G, L-G) merged completely during the PCR. Droplets stabilized with PEG-PFPE₂ surfactant showed substantial merging (**c** depicting post-PCR droplets). Scale bar, 100 μm

maker (Supplementary Fig. 7) to prepare and mix two populations of aqueous droplets: PBS + FITC droplets, containing 2 μM sodium fluorescein salt in PBS; and PBS-only droplets, containing only PBS. Each mixed droplet population was collected in an Eppendorf tube and incubated at 37 °C. Droplets created with H-dTG, M-dTG and L-dTG surfactants were more resistant to dye diffusion than those created with M-G and L-G (note: H-G surfactant was not synthesized as M–G and L–G surfactants did not perform better than dTG-based surfactants), indicating that the dTG head-group improves dye-retention better (Fig. 3 and Supplementary Fig. 8).

We also address the effect of tail length on inter-droplet dye exchange, finding that, for dendritic head groups, droplets produced with the longer chain surfactants are more resistant to leaching than those produced with the short chain (H-dTG = M-dTG > L-dTG; M–G > L–G). Quantitatively, after 24 h incubation, the fluorescence intensity of PEG-PFPE₂ surfactant-stabilized PBS-only drops was, respectively three times and 13 times the intensity measured in PBS-only droplets stabilized by our worst-performing surfactant, L–G (Supplementary Fig. 8c), and our best-performing surfactant, M-dTG (Fig. 3b). Not surprisingly, the fluorescence intensity of M-G surfactant-stabilized PBS-only drops was three times the intensity measured in PBS-only droplets stabilized by M-dTG surfactant (Fig. 3b), confirming the importance of a dense hydrogen bond network at the oil–water interface.

To directly address the contribution of hydrogen-donor activity of the hydroxyl groups in oligo-glycerol-based surfactants, we test if a high salt concentration, known to form ion–dipole interactions and disrupt inter-and intramolecular hydrogen bonding, decreases dye retention in droplets stabilized with the oligo-glycerol-based surfactant M-dTG. Indeed, in presence of 5 M NaCl we see inter-droplet leakage of water-soluble fluorescein dye after 24 h (Supplementary Fig. 9).

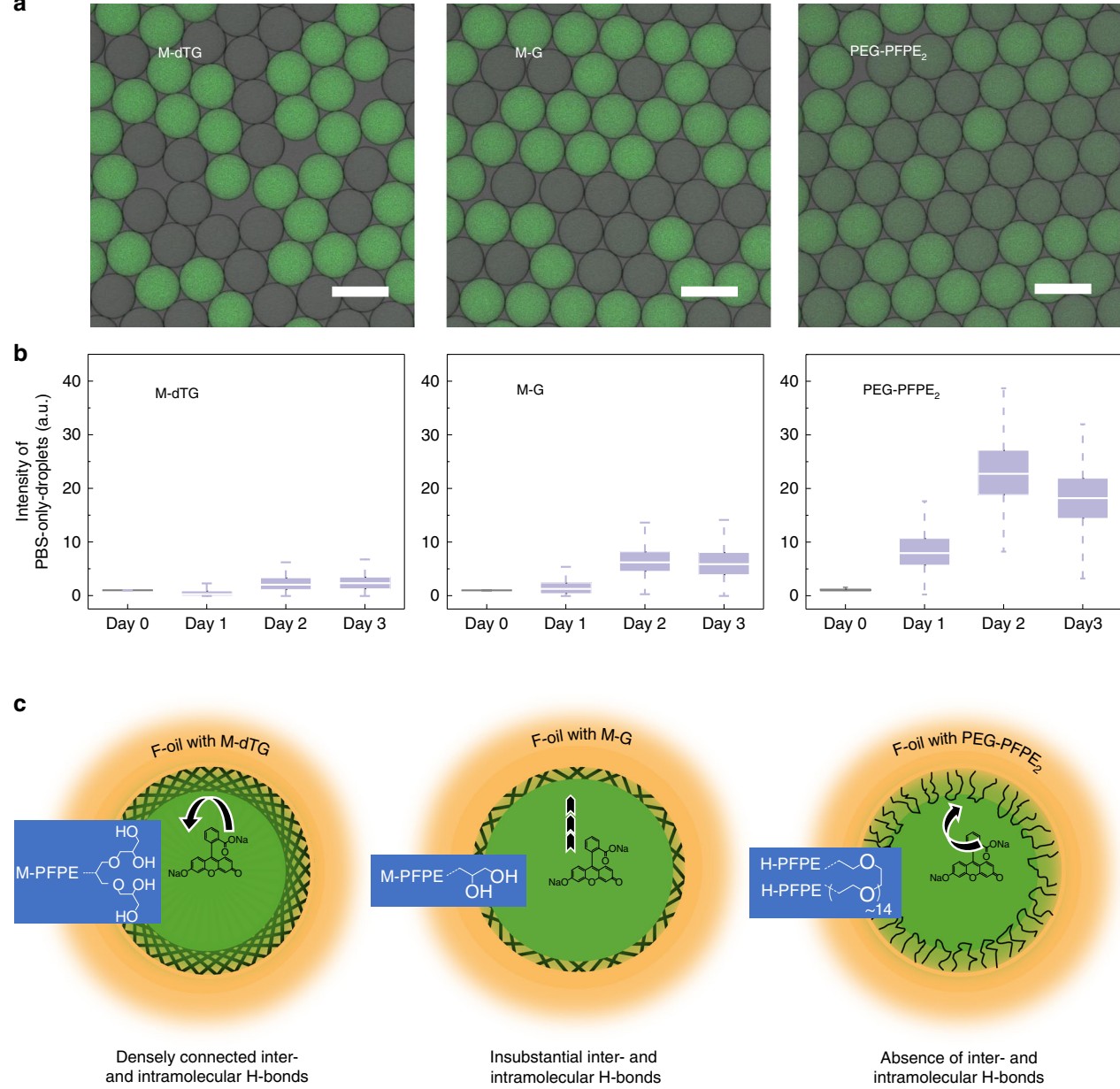

**Fig. 3** Influence of dense hydrogen bond network on inter-droplet diffusion. **a** For each surfactant, we used parallel drop maker to create a mixture comprising equal amounts of PBS-only-droplets and PBS + sodium fluorescein salt-containing droplets. Confocal fluorescent imaging of droplets after 72 h incubation at 37 °C is shown in top panel. **b** We incubated these mixtures, took fluorescence images at the indicated time points, and then performed quantitative analysis of the fluorescence intensity of the 10 randomly selected PBS-only droplets. Box-plot demonstrates that dye was almost completely retained in droplets stabilized by M-dTG, while some transfer was detected in M-G droplets. PEG-PFPE$_2$ surfactant-stabilized droplets showed substantial transfer (left to right). The box plots represent the median (center line), the interquartile range (box) and the non-outlier range (whisker). **c** A model representing the plausible inter- and intramolecular hydrogen bonding from M-dTG, M-G, and PEG-PFPE$_2$ surfactants (left to right). The lines denote hydrogen bonding at the interface of oil and water. For PEG-PFPE$_2$ surfactant, as there is no hydrogen bond donor, there is no inter- and intramolecular hydrogen bond present. Scale bars = 100 µm; a.u. = arbitrary units. Source data of Fig. 3b (middle panel) are provided as a Source Data file

Further, to better define the improved performance of the M-dTG surfactant relative to PEG-based surfactants, we tested retention of the small molecule dye resorufin ($M_w = 235$ g mol$^{-1}$). In our experiments with droplets stabilized with 4% (w/w) PEG-PFPE$_2$ surfactant, almost 50% of the dye is transferred from droplets containing 2 µM resorufin + PBS to PBS-only-droplets after 30 min. However, in droplets stabilized with 2% (w/w) M-dTG surfactant, the inter-droplet transfer of ~50% resorufin dye occurs after 330 min, indicating that inter-droplet transfer between M-dTG stabilized droplets is 11 times slower than between PEG-PFPE$_2$ stabilized droplets (Supplementary Fig. 10).

The high performance of the tri-glycerol dendron-containing (dTG-based) surfactants suggests that the symmetrical geometry, multiple hydrogen bond donors, and flexible ether groups collectively help the dTG polar group form densely connected inter- and intramolecular H-bonds, which act as a strong web-like network at the oil–aqueous interface (see model in Fig. 3, bottom panel). This is supported by our results showing that high-salt concentrations reduce dye retention.

In addition, we believe that the hydrogen-donor activity of dTG-head groups may enhance retention of cargo molecules containing hydrogen-bond accepting atoms, e.g., oxygen (O) and nitrogen (N). When these molecules approach the H-bond-dense water–oil interface, dTG-based surfactants will donate hydrogen bonds to the cargo, increasing its aqueous solubility[28].

G-head groups, due to their limited number of hydroxy groups and lower number of flexible chiral centers, make insufficient inter- and intra-molecular H-bonds and do not significantly interact with cargo containing hydrogen-bond accepting atoms as depicted schematically in Fig. 3c. In contrast, PEG forms neither inter- or intramolecular hydrogen bridges and does not act as hydrogen bond donor to increase dye solubility at the aqueous–oil interface. This may partially explain the relatively poor dye retention of droplets stabilized with the PEG-PFPE$_2$ surfactant. In addition, the PEG-PFPE$_2$ surfactant, as a tri-block copolymer, can form bilayer vesicles to mediate transport between droplets[12,29].

**Cell-based reporter system.** Our findings suggest that M-dTG is the best of the dTG- and G-based surfactants for drop-stability and retention of water-soluble small molecules (fluorescein) and more leaky small molecules (resorufin). To be truly valuable for droplet-based drug-screening, a surfactant must also prevent inter-drop transfer of encapsulated small, non-polar compounds and it should be biocompatible. We developed a cell-based assay, much more sensitive than dye-transfer monitoring, in which reporter gene activity is used to characterize leakage of a drug, doxycycline (DOX), between M-dTG-stabilized droplets. We used a hyperactive piggyBac transposase (hyPBase)[30] and XLone-GFP[31] plasmid constructs to create a Tet-On 3 G DOX-inducible GFP expressing stable cell line, DOX-GFP-HEK 293 (reporter cells) (Supplementary Fig. 11). In the absence of DOX, GFP expression is negligible, whereas 48 h incubation with 500 nM DOX results in easily-detected GFP signal in 73.5% of the cells (Fig. 4a). We then determined that the cell-stream of our parallel drop maker loads cells into droplets roughly according to Poisson prediction (Fig. 4b, c). At five cells per droplet volume, 83% of the droplets contain one or more cells. We further demonstrated that droplet size remains constant over a 72 h incubation (Fig. 4d).

In addition, to test the biocompatibility of the M-dTG surfactant and the PEG-PFPE$_2$ surfactant (a reference standard), we generated cell-containing droplets equivalent to those used for cell culturing experiments (see Fig. 4b, c) and incubated for 24, 48, and 72 h incubation time points. We used perfluoro-octanol to release the cells from the droplets and tested viability using Calcein AM and Ethidium Homodimer-1 dyes based live/dead cell viability assay kit (Invitrogen). At each time point, cell survival is roughly 85–90% (Fig. 4e). By comparison, the cell survival is about 95% when cells were cultured under standard conditions in well plates. This demonstrates that dTG-based surfactant, M-dTG, is not cytotoxic to cells and is suitable for cell-based assays.

We used the DOX-GFP-HEK 293 cells to test DOX transfer between droplets within populations stabilized by either M-dTG or PEG-PFPE$_2$ surfactant. We used the surfactants at roughly equimolar concentrations, 4% PEG-PFPE$_2$ and 2% M-dTG. We created a mixture of cell-containing droplets and droplets containing 1 µM DOX, incubated for the indicated times, and then released the cells from the droplets. In positive control sample, cells were encapsulated and cultured in droplets with DOX at the indicated concentrations. To compare GFP expression level of cells grown in droplets with GFP expression level of cells grown in bulk (standard culture plates), we cultured cells in six-well plates. When cells were cultured in bulk with DOX

(positive control in bulk), almost 60% of them were GFP+ after day 1, 73.5%, were GFP+ after day 2, and 69% after day 3. In-droplet control experiments showed that cell response to DOX is roughly equivalent in droplets stabilized with M-dTG or PEG-PFPE$_2$ surfactant. After 1 day a high percentage of cells encapsulated with 500 nM DOX, 52.3% and 53.9% respectively, were GFP+; 31.3% and 25.5%, respectively, were GFP+ when incubated with 200 nM DOX (Fig. 5 and Supplementary Fig. 12).

In the experiments to test transfer of DOX from DOX-only droplets (1 µM) to cell-only droplets, we see that GFP expression in droplets stabilized with M-dTG is lower than in droplets stabilized with PEG-PFPE$_2$, 37.1% vs 44.5% (compare Fig. 5 column 4 with supplementary Fig. 12 column 3). When cells incubated for one day are compared, the inter-droplet transfer from M-dTG stabilized droplets containing 1 µM DOX induces cells to express GFP at roughly the same level as seen for cells incubated for one day in control droplets containing 200 nM DOX. Our results also show that, for a given concentration of DOX, cells incubated in bulk conditions yield more GFP+ cells than cells incubated in droplets (Fig. 5, compare columns 2 and 3). This is not surprising, as the effective concentration of cells cultured in droplets is ~10 times that of cells cultured in standard conditions.

These studies of inter-droplet transfer clearly suggest that M-dTG-stabilized droplets, relative to those stabilized with PEG surfactant, are significantly better at retaining the water-soluble dye fluorescein sodium salt, and are measurably superior at retaining other small molecules such as resorufin and doxycycline. Thus, dTG-based surfactants, which are synthesized from easily-sourced building blocks, have great potential as robust and economical droplet stabilizers and will be powerful reagents for droplet-based single cell analysis even under PCR conditions and for drug screening applications.

In addition, we provide insights into surfactant design through systematic analysis of two different polar head groups with three fluorinated tail groups. We found that polar groups can interact through inter- and intramolecular hydrogen bonds and provide the emulsified droplets high-temperature tolerance and high retention of encapsulated small molecules. Further, the hydroxy moieties of the polar head groups can easily be functionalized, enabling surfactant optimization for specific chemical environments. We expect our surfactants to be of immediate use in droplet-based experiments requiring high droplet integrity.

## Methods

**Materials**. We purchase the monocarboxylic acid-terminated perfluoropolyethers, brand name Krytox, of three different chain lengths that are Krytox 157-FSH (M$_w$ = 7000–7500 g mol$^{-1}$), Krytox 157-FSM (M$_w$ = 3500–4000 g mol$^{-1}$), and Krytox 157-FSL (M$_w$ = 2000–2500 g mol$^{-1}$) from LUB SERVICE GmbH (Germany). We buy HFE 7100 and HFE 7500 oils from 3 M. We obtain 2,2-Dimethyl-1,3-dioxolane-4-methanamine (acetal-protected mono-glycerol-NH$_2$, G-NH$_2$) from Merck (Germany). We purchase tri-block copolymer fluorosurfactant PEG-PFPE$_2$ (EA surfactant) from RAN Biotechnologies (Beverly, MA). The EA surfactant has two perfluoropolyether tails (each having a MW ~6000 g mol$^{-1}$) coupled to a homo-bifunctional PEG600-amine head group[1]. All other chemicals we purchase are reagent grade. These chemicals are either from Acros Organics (Belgium) or from Merck (Germany) unless otherwise stated. They are used as received without further purification. All moisture sensitive reactions are conducted in flame-dried glassware under dry conditions. The acetal-protected triglycerol dendron-amine (dTG-NH$_2$) is synthesized following previously reported synthetic routes with slight modifications indicated below[15,32]. For the fluorosurfactants' synthesis the protocol by Holtze et al.[11], Baret et al.[12], and Wagner et al.[15], is used. The surfactant solution is prepared in the HFE-7500 oil, a biocompatible oil phase, at 2% and 4% by weight. To make microgels, two polymer precursors, homo-bifunctional polyethylene glycol-cyclooctyne (PEG-DIC) and dendritic polyglycerol sulfate azide (dPGS-N$_3$) are reacted according to previously reported synthetic schemes[33].

**Analytical methods**. NMR spectra are recorded on an ECX400 spectrometer (Jeol Ltd., Japan), or an AMX 500 spectrometer (Bruker, Switzerland). Proton NMR chemical shifts are reported as δ values in ppm. Deuterated solvent peak is used to

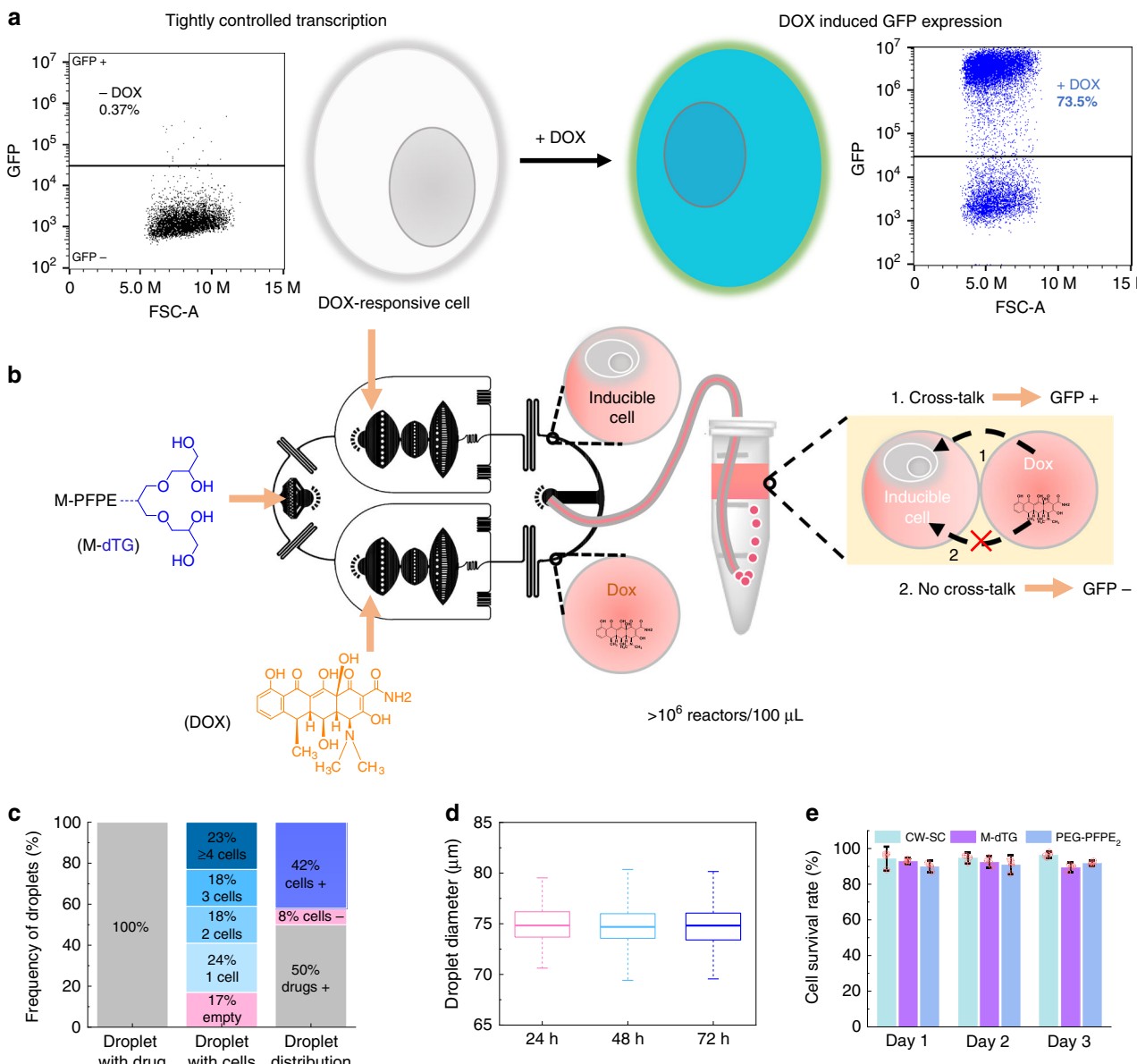

**Fig. 4** Cell-based reporter system to test inter-droplet drug diffusion. **a** HEK 293 cells stably transfected with a doxycycline responsive GFP reporter construct (reporter cells) show no GFP expression in the absence of drugs and become highly fluorescent after 48 h incubation in the presence of 500 nM DOX-solution. **b** Homogeneous population of DOX-containing droplets and Reporter-cell-containing droplets was generated using a parallel drop maker that creates an equal number of droplets from two independent aqueous streams. **c** Cell-loading into droplets roughly follows Poisson predictions. We imaged 533 droplets immediately after cell encapsulation to determine cell occupancy. We used an input density of five cells per droplet volume and found that 83% of the droplets contained one or more cells. Therefore, with cells at this concentration, the parallel drop maker generates a mixed droplet population in which 42% droplets contain one or more cells, 50% contain only doxycycline, and 8% are empty. **d** Box-plot of droplet size distribution after 24, 48, and 72 h incubation at 37 °C demonstrates that the mean average droplet diameter remained constant (~75 μm) over the time course ($n =$ 204–226 droplets for each time point). The box plots represent the median (center line), the interquartile range (box) and mean ± 1.5 x s.d. (whisker). **e** Viability of cells cultured in wells or in droplets. As a positive control we cultured cells at standard concentration in culture wells (CW–SC) ($1 \times 10^6$ cells/ ml). We used M-dTG and PEG-PFPE$_2$ surfactants to generate cell encapsulated droplets and incubated droplets at 37 °C for the indicated times. We isolated cells from the droplets to perform live/dead assay using Calcein AM and Ethidium Homodimer-1 dyes based live/dead cell viability assay kit (Invitrogen). We counted ~500–2000 cells to estimate the cell survival rate at the indicted time points. Data are presented as mean ± s.d., $n = 4$ images from distinct areas. Source data of c–e are provided as a Source Data file

calibrate the recorded peak. To record IR spectra, Nicolet AVATAR 320 FT-IR 5 SXC (Thermo Fisher Scientific, USA) is used with a DTGS detector from 4000 to 650 cm$^{-1}$ wavenumbers. Bright field and fluorescence imaging are obtained with Zeiss Axio Observer (Germany) and Leica confocal microscope (TCS SP8, Germany). To record green fluorescence signals, 488 nm excitation wavelength is used, and the emission signals are detected by a 520/55 nm band pass filter. A high-speed Phantom MIRO ex2 camera (Vision Research, USA) is employed for brightfield imaging during microfluidic droplet preparation. FACSAriaIII (USA) cell sorter and BD Accuri™ C6 Plus (USA) flow cytometer are used for cell sorting

and to analyze the GFP expressing cells, respectively. Flow cytometry data is analyzed using FlowJo v10. Homemade MATLAB scripts are used to analyze droplet size distribution by finding droplet circles and extracting the corresponding diameters that are adjusted to final diameter according to the actual pixel size of the image. OriginPro 2019b (Academic) was used to prepare Boxplots and bar charts.

**Synthesis of acetal-protected dTG-NH₂.** Acetal-protected tri-glycerol dendron-hydroxy (dTG-OH) is dried under high vacuum at 60 °C overnight. Thereafter,

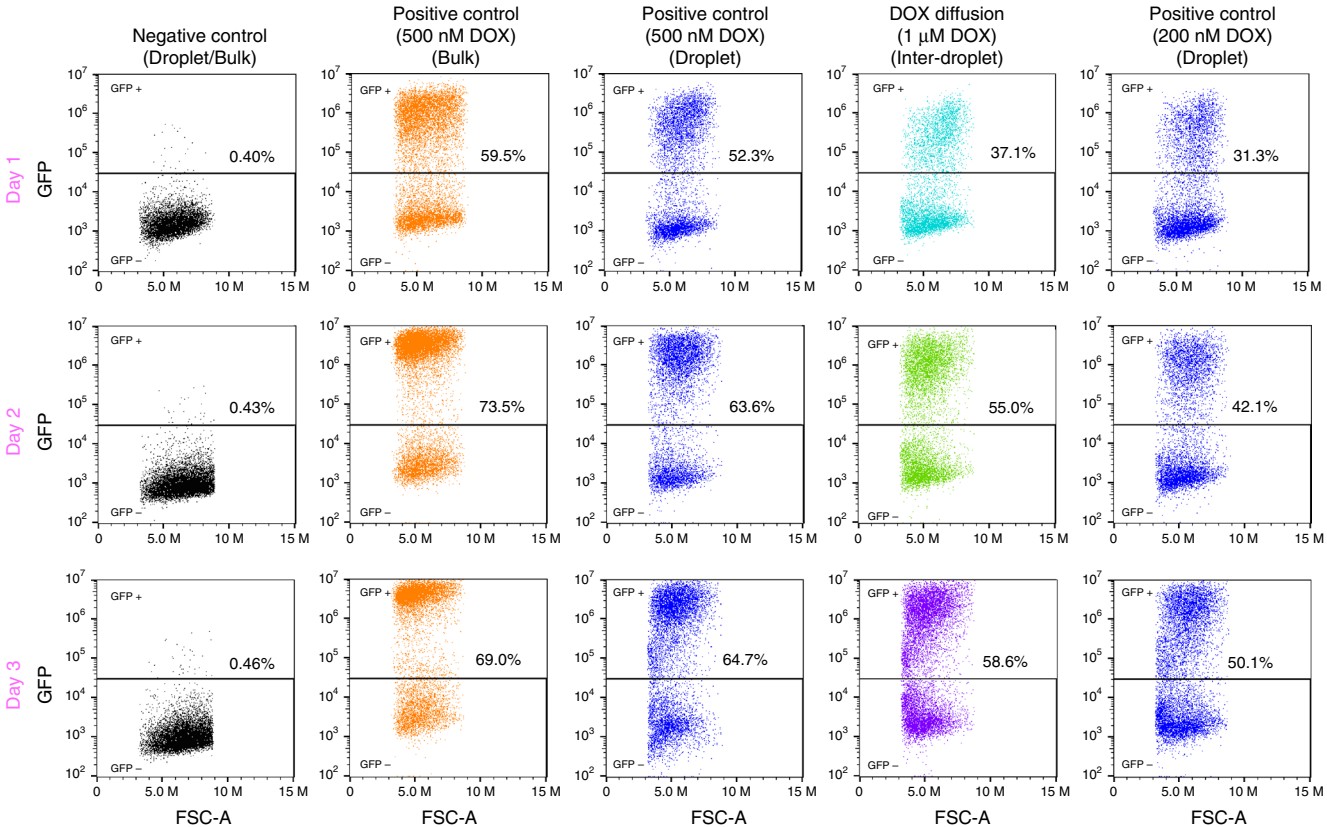

**Fig. 5** DOX-inducible GFP-reporter cells to quantify drug transfer. We used the parallel drop maker to generate homogenous mixtures in which 50% of the droplets contained DOX (at the indicated concentrations), 42% contained ≥1 cell, and 8% were empty. We incubated droplets at 37 °C for the indicated times, isolated the cells from the droplets, and quantified the GFP+ cells using flow cytometer. GFP intensity is plotted against forward scatter-area (FSC-A). For the positive control in bulk, we cultured the DOX-GFP-HEK 293 cells with DOX in six well plates using standard cell culture methods. For the Positive Control (Droplet), all droplets, including those with cells, contained DOX at the indicated concentration

dTG-NH₂ is prepared in a three-step process. In the first step, the hydroxy groups are mesylated using 1.5 equivalents of methane sulfonyl chloride (MsCl) and 2 equivalents of triethyl amine. This reaction is performed in DCM at 0 °C to RT overnight under inert conditions while stirring. The mesylated acetal-protected tri-glycerol (dTG-OMs) is extracted in dichloromethane (DCM) against water, the solution is dried over $Na_2SO_4$ salt, and then the volume is concentrated under reduced pressure. To substitute the mesyl group by azide, dTG-OMs in dimethylformamide (DMF) is dissolved and $NaN_3$ salt (3 eq.) is added into it. The reaction mixture is heated with continuous stirring for 3 days at 75 °C under argon. Then the solvent is evaporated, and the compound is extracted in DCM followed by drying over $Na_2SO_4$ salt, and filtering through cotton wool. After removing the solvent, the acetal-protected dTG-N₃ is obtained in quantitative conversion of mesyl to azide groups. Finally, to convert azide to amine functionalities, 10% (w/w) a palladium catalyst (10% Pd on Charcoal) is used with respect to the weight of acetal-protected dTG-N₃ for hydrogenation. The substrate is dissolved in dry ethanol in a hydrogen reactor and pressurized to ~5 bar hydrogen atmosphere at RT for 3 days under vigorous stirring. Then the Pd/Charcoal is filtered using a well-packed bed of cellite®545 on a fritted glass filter (pore size 4). After concentrating the solution, the acetal-protected dTG-NH₂ is obtained in almost quantitative yield.

**Synthesis of di-block dendronized surfactants**. The diblock-dendronized fluorosurfactants are synthesized in a three-step process. In the first step, oxalyl chloride is used to activate the acid group in Krytox, producing acyl chloride terminated Krytox[11,12,15]. The reaction is performed in a one-neck round bottom Schlenk flask equipped with a magnetic stirrer bar. Krytox (1 eq.) is dried at 100 °C under HV for 3 h and after cooling down the flask to RT, it is dissolved in dry HFE7100. Then a catalytic amount of DMF is injected into the Krytox solution under argon atmosphere. Then oxalyl chloride (10 eq.) is added and the formed toxic gases are removed via a Schlenk line. After 30 min, the reaction is continued overnight at RT under argon atmosphere with vigorous stirring. Then all the volatile gases and the solvent are removed under high vacuum (HV) using an additional cold trap. Krytox 157 FSH, Krytox 157 FSM, and Krytox 157 FSL as high, medium, and low molecular weight fluorinated blocks, respectively are used in the same way for all three types. In the second step, 1.3 equivalents of dry

acetal-protected G-NH₂ or dTG-NH₂ are used to react with the activated-Krytox. DCM is used to dissolve the acetal-protected oligo-glycerol-amine precursor and then added dropwise into the activated-Krytox solution in HFE7100 under argon atmosphere. Subsequently triethylamine (3 eq.) is added as an organic base into the reaction mixture. For a better miscibility of these different solvents, a ratio of 1:1 or 1:0.75 of DCM and HFE7100 is used. Then the reaction mixture is refluxed for 2 days at 50 °C and the crude reaction mixture transferred into a one-neck round bottom flask to remove the solvents and acid vapors under reduced pressure. The dried compound is dissolved in HFE7100 and methanol and stirred vigorously for ~20 min. HFE7100 has a density of 1.52 g/ml, whereas methanol has a density of 0.79 g/ml. Due to such a high-density gradient the two solvents phase separate. Then the top clear organic phase of methanol (10×) is removed. This washing step removes the base, salt, and the excess unreacted dendron molecules, providing purely the di-block dendronized fluorosurfactants. Finally, acidic methanol, 0.5–1.0 ml of 37% (w/w) HCl solution in 20 ml of methanol for a 10 g batch, is added into the acetal-protected oligo-glycerol containing fluorosurfactant, dissolved in HFE7100 to deprotect the acetal groups from the oligo-glycerol molecules overnight at 50 °C, generating water soluble polar group in the fluorosurfactant. After repeated washing with methanol (20×), the different fluorosurfactants are obtained in 75–85% isolated yield after HV drying.

**PDMS device fabrication**. The PDMS microfluidic devices are fabricated according to the methods described in Mazutis et al.[34], with minor modifications and details specific to our experiments. SU8-on-Si wafer is prepared using spin coated SU-8 2050 photoresist (MicroChem, MA) layer of a desired thickness that is UV etched using the photomask. PDMS is baked at 65 °C for 3 h after pouring onto the SU-8 master. To remove the debris after hole punching, we sonicate the PDMS slab in an isopropanol bath, dry with pressurized air, then bake at 75 °C to ensure all isopropanol is evaporated. After plasma bonding of PDMS slab to glass, the PDMS-on-glass device is heated on a 75 °C hot plate for ~10 min and then incubated at 65 °C overnight in an oven to enhance bonding. To make the channel surfaces hydrophobic, PTFE-syringe-filtered Aquapel (PPG Industries) is injected into the channel, incubated ~60 s at room temperature, and then blown out of the channels using pressurized nitrogen. To remove traces of Aquapel, PDMS device is incubated at 60 °C for ~2 h.

**PCR experiments**. Microfluidic drop making devices are used to create ~95 μm diameter monodisperse droplets stabilized by the indicated surfactants (2% w/w) in HFE7500 carrier oil. Each emulsion of droplets is created from 40 μl PCR mix comprising 25.2 μl water, 8 μl 5x Phusion HF detergent-free Buffer (F520L, Thermo Fisher), 0.8 μl 10 mM dNTPs (diluted from 25 mM dNTP mix, Thermo Fisher, R1121), 1.6 μl 2 μM forward primer (5′ TCGTCGGCAGCGTCAGATGTG 3′, ordered from IDT), 1.6 μl 10 μM reverse primer (5′ GTCTCGTGGGCTCGGAG ATGT 3′, ordered from IDT), 0.4 μl Phusion High-Fidelity DNA Polymerase (F530L, Thermo Fisher), 0.4 μl 20 mg/mL bovine serum albumin (BSA, B14, Thermofisher), 0.4 μl 10% tween-20 (diluted from Tween-20, Sigma-Aldrich, P9416-50 mL), and 1.6 μl template (4 μl 2.5 μg/mL Lambda Phage genome is fragmented and tagged by 6 μl reagents from Nextera kit from Illumina following its protocol, FC-121-1031, then diluted 100 times by water). The PCR program is 98 °C for 30 s; then 35 cycles of 98 °C for 7 s, 60 °C for 30 s, and 72 °C for 20 s; then a final step of 72 °C for 10 min. After PCR thermocycling, droplets are broken by adding perfluorooctanol (PFO; 370533, Sigma) to the droplets (five volumes of PFO to one volume of the droplet aqueous contents). 5 μl aqueous phase of each sample is electrophoresed on a 2% agarose using 1x TAE buffer. In total 6 μl GeneRuler 100 bp DNA ladder (Thermo Fisher, SM0241) is used as reference. Gel is stained post-electrophoresis with gel red (41002, Biotium) and imaged using a UV light box and camera (Supplementary Fig. 5c).

**Release of microgel particles**. After droplet preparation, the emulsion droplets are incubated at 37 °C for ~30–60 min to allow strain-promoted azide-alkyne cycloaddition (SPAAC) reaction generating cross-linked networks. To release the microgel templates, we first pipet to remove excess oil underneath the droplet emulsion and then the droplet emulsion is washed 5–10 times. For each wash, a volume of HFE7500 oil equivalent to five times that of the droplet emulsion is added to the droplets and mixed by gentle inversion of the Eppendorf tube. This oil is then removed from underneath the droplets. After washing, a volume of PBS medium equivalent to five times that of the droplet emulsion is added to the droplets. This causes the droplets to coalesce during gentle inversion and release the microgel particles into the access aqueous phase.

**Parallel drop maker**. The parallel flow-focusing drop maker uses one oil stream and two aqueous inlets to create two distinct populations of same-size droplets that are mixed in incubation line before reaching the outlet of the channel and then collected into an Eppendorf tube (Supplementary Fig. 7). The flow-focusing nozzle is 55 μm × 55 μm. Flow rates of 1200 μl/h for the HFE7500 continuous phase oil and 300 μl/h for each of the two aqueous streams creates droplets of ~90 μm diameter. Syringe pumps (Harvard Apparatus, USA) are used to control the flow of different liquid streams.

**Dye diffusion experiment**. A parallel drop maker (Supplementary Fig. 7) is used to prepare two populations of ~90 μm diameter droplets, which are mixed on-chip before collection in an Eppendorf tube. One droplet population contains PBS + sodium fluorescein dye and the other droplet population contains only PBS. The mixed droplet population is incubated at 37 °C and imaged by microscopy at days 0, 1, 2, and 3. The green fluorescence intensity of the PBS-only-droplets is quantified using the line profile tool of Leica software. The mean fluorescence value of day zero PBS-only droplets was assumed to be the background and was subtracted from the day 1, 2, and 3 measurements.

**DOX-inducible HEK293 stable cell line generation**. Human embryonic kidney 293 cells (HEK293, passage#6, ATCC® CRL-1573™) are cultured in Dulbecco's modified Eagle's medium (DMEM) under standard cell culture conditions. The growth medium is supplemented with 15% (v/v) fetal bovine serum (FBS) and 1% (v/v) penicillin-streptomycin (P-S). To generate a stable Tet-On 3 G DOX-inducible green fluorescence protein (GFP)-expressing reporter-cell line, HEK-293 cells are transfected with hyperactive piggyBac transposase (pCMV-hyPBase, a kind gift from the Sanger Institute, UK)[30] and XLone-GFP (a gift from Xiaojun Lian; Addgene plasmid # 96930)[31] plasmid constructs (Supplementary Fig. 11). Lipofectamine 3000 (Thermofisher) is used to transfect the plasmids, 5 μg of the XLone-GFP and 2.5 μg of the pCMV-hyPBase, into the cells in 6-well plates according to the supplier's instructions. After 72 h of transfection, Blasticidin S is used at 10 μg/ml in DMEM with 5% FBS and 1% P-S for 10 days to select the stably transfected cells. Cells are treated with 4 μM DOX for 24 h to turn on the GFP expression and then FACS is used to isolate a pure population of cells having maximum GFP fluorescence intensity. Stable monoclonal cell lines of high DOX sensitivity were generated using limiting dilution method and expanded under Blasticidine S selection. Cells are maintained with selection pressure for all downstream experiments. Reporter-cells are repeatedly tested for mycoplasma by PCR.

**Flow cytometry analysis for inducible GFP expression**. Prior to flow cytometry analysis, encapsulated cells are released from the emulsion droplets. Initially, excess oil underneath the droplet emulsion is removed by pipetting and then the droplet emulsion is washed five times. For each wash, a volume of HFE7500 oil equivalent to five times that of the droplet emulsion is added to the droplets and mixed by gentle inversion of the Eppendorf tube. This oil is then removed from underneath

the droplets. After washing, volume of 20% PFO in HFE7500 oil equivalent to five times that of the droplet emulsion is then added and mixed by gentle inversion. This causes the droplet to coalesce into a single aqueous fraction. Fresh HFE7500, equivalent to five times the aqueous volume is then added to dilute the remaining PFO. This HFE7500/PFO mixture is removed and then three aqueous volumes of DPBS, containing 0.5 mM EDTA and pre-heated to 37 °C, is added to the cell-containing medium to allow the cells dissociate into single cells for 5 min. The cell suspension is gently mixed by pipetting to fully disperse cells and immediately used to quantify the GFP expression by flow cytometer. FlowJo is used to analyze these recorded data. Flow cytometer gating is performed based on corresponding DOX-untreated control reporter-cell population. Singlet cell population gating and a comparison of GFP expression in presence and absence of DOX are displayed in the Supplementary Fig. 13.

**Reporting summary**. Further information on research design is available in the Nature Research Reporting Summary linked to this article.

## Data availability
Data supporting the findings of this work are available within the paper and its Supplementary Information files. A reporting summary for this Article is available as a Supplementary Information file. The datasets generated and analyzed during the current study are available from the corresponding author upon request. The source data underlying Figs. 3b (middle panel), 4c, d, e and Supplementary Figs. 4a–e, 5c, and 8a–c are provided as a Source Data file.

## Code availability
The homemade MATLAB scripts are available from the corresponding author upon reasonable request.

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

## Acknowledgements

We thank Katharina Goltsche for acetal-protected tri-glycerol dendron hydroxy (dTG-OH) synthesis, Dr. Roy Ziblat for MATLAB script, Dr. Badri Parshad and Dr. Abhishek Kumar Singh for discussions, Yong Hou for helping in live/dead assay. This work was supported by a Dahlem Research School (DRS) grant, the core-facility Biosupramol (www.biosupramol.de) to R.H., the NSF (DMR-1708729) and the Harvard Materials Research Science and Engineering Center (MRSEC) (DMR-1420570). The work was also funded by the Deutsche Forschungsgemeinschaft (DFG, German Research Foundation) – project id 387284271 – SFB 1349 Fluorine-Specific Interactions.

## Author contributions

M.S.C. designed the research, analyzed the data and wrote the manuscript; W.Z. performed the PCR experiment and analyzed the data. S.K. and M.S.C. jointly contributed to scale up of DNA plasmid constructs and worked on transfected stable cell lines generation. J.H. analyzed the data and contributed to the writing. X.Z. assisted in writing. P.D. synthesized polymer precursors for microgel template preparation. D.W. and R.H. supervised the study, analyzed the data and wrote the manuscript.

## Competing interests

Freie Universität Berlin and Harvard University have filed an international patent (PCT/EP2018070036) on these dendronized fluorosurfactants. M.S.C., R.H. and D.W. are affiliated with these organizations and the patent. The remaining authors declare no competing interests.
