## [Peer Review File · Nature Communications]

Reviewers' comments:

Reviewer #1 (Remarks to the Author):

The authors report a new type of fluorosurfactant with a dendronized polyol head group with improved microdroplet stability upon PCR thermal cycling and prevention of inter-droplet transfer of small molecules such as fluorescent dyes and drugs.

The development of a novel, better performing fluorosurfactant is crucial for advancing the field of droplet microfluidics and will have a positive impact to a wide range of applications. And the two features that the authors chose to demonstrate (that is, thermal stability and inter-droplet transfer) are the key parameters for enabling biological assays in microdroplets.

This reviewer thinks that the experiments are well designed and executed and the manuscript is clearly written. However, there are a few claims that need clarification and/or more supporting evidence. I would like to see the following points are addressed satisfactorily with a revised manuscript before I recommend publication in Nature Communications.

1. In Introduction, the authors claim that "Di-block surfactants do not suffer from these problems" because the polar head group can be added to the reaction in an excess and removed by simple purification. The uncertainty in molecular weights of PEG and PFPE does contribute to the problem, but the main contrast between the tri-block PEG-PFPE2 and the authors' surfactant seems to be the polar nature of the head group facilitating easier purification. The unique solubility profile and the high viscosity of fluorinated polymer PFPE is the main challenge for purification in fluorosurfactant synthesis. More explanation on your purification scheme (phase separation and solubility) will strengthen your argument.

2. Have you quantified the actual purity of your di-block surfactant? I only found the yield of your synthesis in the manuscript but not purity. How would you measure the purity of your final product?

3. As the authors mention, tri-block PEG-PFPE2 surfactant will be a mixture of di-block and tri-block polymers due to incomplete reaction and uncertainty in molar ratio calculation. However, mixed surfactant system can often perform better in terms of droplet stability. Also, tri-block surfactant will have a denser polymer brush dissolved in oil phase, thus providing stronger barriers between droplets, all of which may explain the success of PEG-PFPE2 surfactant. Why does a di-block polymer with a smaller head group and a smaller tail group perform better?

4. The authors mention, "We created droplets using these surfactants at roughly equimolar amounts, corresponding to 2% w/w H-dTG and 0.7% w/w L-dTG. Under these conditions, H-dTG droplets were more stable than L-dTG-stabilized droplets, suggesting that the H-dTG longer fluorinated tails can provide more stability than shorter tails (data not shown). Of note, droplets made with 2% w/w L-dTG were more stable than those formed with the PEG-PFPE2 surfactant (Fig. 2c)." This reviewer thinks that comparing surfactant performance with samples containing equimolar surfactant concentration is a fairer comparison. It appears that 2% w/w final surfactant concentration is simply an experimental choice (a concentration that does not affect viscosity of oil phase too much) and can't serve as a criteria for comparing surfactant performance. The optimal surfactant concentration can be different for each type of surfactant. Based on estimated molecular weights (15 kD for PEG-PFPE2, 7.5 kD for di-block H, 4 kD for di-block M, 2.5 kD for di-block L), the molar concentration can vary as large as 6 fold (e.g., between PEG-PFPE2 and di-block L) for the same w/w percentage concentration. I suggest that the data on emulsion stability under PCR condition at equimolar concentration (corresponding to 5% w/w PEG-PFPE2 concentration) be included in the revised manuscript.

5. Have you tried Taqman-type PCR with your surfactant? The probe generates fluorescent dye molecules as a result of polymerase activity and therefore such an in-droplet reaction and

fluorescence imaging before and after PCR would be a great demonstration of both droplet stability and inter-droplet dye transfer rate.

6. You mentioned that hydrogel droplets are generated at kHz rates (Supplementary Figure 5) and gels were transferred to aqueous solution. Making droplets of highly viscous polymer at a high speed is challenging, but this reviewer could not find any information on the device and flow rates used, nor the work-up procedure for recovering hydrogel particles. Please add more detailed information, which would be useful to the interested readers.

7. Even though the choice of fluorescein and doxycycline makes sense, this reviewer disagrees with labeling them as water-soluble and DMSO-soluble. Fluorescein is soluble both in water and DMSO. The same is true for doxycycline. According to the literature (<https://www.ncbi.nlm.nih.gov/pubmed/34018> and <https://pubs.acs.org/doi/abs/10.1021/ie060055v>), doxycycline and its more well-known cousin tetracycline dissolve in water fairly well (up to ~40 mM). It is common practice to dissolve antibiotics or inducers in DMSO at high concentration and dilute to culture media, but such labeling might give a false impression that fluorescein and doxycycline are representative of hydrophilic and hydrophobic molecules.

8. Fluorescein is a polar molecule with a phenol and a carboxy group and doxycycline has 5 hydroxy groups and one amide bond, which makes them efficient participants to hydrogen bonding network. This may limit the applicability of the surfactant to hydrogen bond-capable molecules. Fluorescence imaging data of inter-droplet transfer for more leaky dyes such as resorufin and Nile Red would be informative and better define the performance gains and applicability of the dendronized di-block surfactant.

9. How sensitive is the GFP reporter system within droplets? 90% cell survival rate in droplet culture does not mean that their GFP protein synthesis is as efficient as the bulk cell culture. I could not find the data showing the sensitivity/efficiency of GFP induction when the doxycycline is present in the droplet. Preferably, the GFP-positive fraction of cells should be compared between bulk culture and droplet culture at the same doxycycline concentrations to ensure that the effect shown in Figure 5 (for the DOX diffusion column) is mainly due to prevention of inter-droplet transfer, rather than the result of direct impediment of GFP induction by droplet environment.

10. In Figure 3b, why would the average intensity go down from Day 2 to Day 3 for PEG-PFPE2 sample? Shouldn't the fluorescence intensity of PBS-only droplets go up until it saturates to the maximum value (theoretically, half of the intensity from the initial dye-containing droplets)? Due to high variability in fluorescence intensity in the droplet population, it may be challenging to distinguish PBS-only droplet and dye-containing droplet. What were the selection criteria for image analysis? Were there two distinct populations over the course of experiments?

Reviewer #2 (Remarks to the Author):

General comments:

This manuscript presents a dendritic oligo-glycerol-based surfactant to demonstrate that the degree of inter- and intra-molecular hydrogen bonds and the dendritic configuration may modulate the stability of droplets in terms of thermal stability and inter-droplet transfer. The presented surfactant is considered of interest to the field, however, the provided data is insufficient to fully support the claims.

Specific comments:

- Hypothesis to be validated:

(1) The authors stated that the length of fluorinated tails would alter the droplet stability (Figure 2, and in the paragraph stating that the H-dTG can provide more stability than L-dTG). However, the

presented data is confusing. If the reviewer has gathered the information correctly, the surfactant molecules on the oil-drop interface matters, given that the H-dTG droplets were observed more stable than the L-dTG one at the equimolecular condition?

Yet, moving on to the description of Figure 3, the best performing group became M-dTG? If this was determined by the inter-droplets leakage, the data for H-dTG and L-dTG shall be presented as a comparison as well.

(2) The authors claimed that the improved thermal stability and minimized inter-droplet transfer of molecules (FITC and doxycycline) of the dTG droplets may be attributed to the inter- and intra-molecular hydrogen bonds. Any direct evidence to prove it? For example, oxidize the hydroxyl groups to carboxyl group to reduce the strength of hydrogen bonding, or shield the inter- and intramolecular hydrogen bonding in H-dTG, MH-dTG and L-dTG by some small molecules.

(3) In line with the previous point, the length of fluorinated tails of the PEG600-based surfactant shall be clearly described. Is it H-PFPE as illustrated in Figure 2b?

Further, was the concentration of PEG-PFPE2 also 2% w/w throughout the manuscript? The equimolar counterpart shall be contrasted with dTG-surfactants for a fair comparison.

The PEG-PFPE2 based surfactant is commercially available by different vendors. Even the same vendor, for example ShpereFluidics, there are different kinds. Which commercially available surfactant has this study employed?

(4) The droplet stability may be intuitively understood by the bright field images as shown in Figure 2 or Supplementary Figure 3. But it may provide a quantitative understanding by the variation of droplet diameters as shown in Figure 4d. Also, how many droplets that the study have investigated for each condition?

(5) Proper statistical analysis shall be conducted to validate the difference as claimed.

- Inter-droplet DOX transfer:

(1) If the minimized inter-droplet transfer of dTG-based surfactant is due to the hydroxyl groups, the data presented with water-soluble fluorescein makes sense. But it may not be necessary true for the DMSO soluble or hydrophobic drugs such as DOX. There is only data of M-dTG droplets but not the PEG-PFPE2 ones. The comparison of the two types of surfactant for their inter-droplet DOX transfer is considered necessary.

(2) The authors claimed that HEK 293 cell survival was above 90% in single emulsions for 3 days culture, which is quite surprising. It is recommended to test the cell viability with dual fluorescence staining such as Calcein-AM and PI, as Trypan Blue staining result is known to be not as accurate, and may be affected by the staining protocols.

- Has the interfacial tension with the dTG-based surfactant been characterized, since interfacial tension would also interfere the droplet size and stability?

Response to Referees:

We would like to thank both the reviewers for their excellent suggestions, comments and questions. Most importantly, we highly appreciate that they found our work is interesting and has potential to advance the field of droplet-based microfluidics.

Reviewer #1:

1. In Introduction, the authors claim that "Di-block surfactants do not suffer from these problems" because the polar head group can be added to the reaction in an excess and removed by simple purification. The uncertainty in molecular weights of PEG and PFPE does contribute to the problem, but the main contrast between the tri-block PEG-PFPE₂ and the authors' surfactant seems to be the polar nature of the head group facilitating easier purification. The unique solubility profile and the high viscosity of fluorinated polymer PFPE is the main challenge for purification in fluorosurfactant synthesis. More explanation on your purification scheme (phase separation and solubility) will strengthen your argument.

Response:

In our work, the oligo-glycerol derivatives in their protected form are only soluble in different organic solvents such as dichloromethane (DCM), methanol, THF etc. In addition, without vigorous stirring, these organic solvents phase separate when mixed with fluorinated solvents, including HFE7100 and HFE7500, due to their low-density profiles. Taking advantage of this density gradient between a fluorinated solvent and a non-fluorinated solvent, the excess protected oligo-glycerol moiety can easily be removed by washing with excess of DCM. Further, washing of the deprotected fluorosurfactant with THF or methanol results in pure di-block fluorosurfactant with no undesired products. More broadly, when the carboxylic acid group at the Krytox terminus is activated with an excess of oxalyl chloride, its reactivity towards a nucleophile is high. Certainly, this reactivity can be used to our advantage when an excess of an amine-functionalized polar group is employed to create purely di-block copolymer surfactant provided that the reaction is carried out under argon condition. Thus, it is possible to synthesize the fluorosurfactant without unreacted Krytox block. In addition, taking advantage of the polar nature of the tri-glycerol dendron in its protected and deprotected forms, we can remove the excess of it including triethylamine salt during repeated post-synthesis washing. The abovementioned explicit explanation on our purification route strengthens our argument and are mentioned briefly on the revised manuscript on page 4.

2. Have you quantified the actual purity of your di-block surfactant? I only found the yield of your synthesis in the manuscript but not purity. How would you measure the purity of your final product?

Response:

Since the unreacted Krytox can interfere with PCR experiments and facilitate charge interaction mediated solute exchange, primarily this is considered an impurity in the final surfactant. Undoubtedly, this is more likely when synthesis of a tri-block copolymer fluorosurfactant is desired. With this in mind, to measure the purity of our final product, we check if there is any free

carboxylic acid stretch present in the FTIR spectra at 1775 cm^{-1} . When the amidation reaction goes 100%, the carboxylic acid stretch shifts to $1740\text{-}1710\text{ cm}^{-1}$ in FT-IR spectroscopy due to amide (-NH-CO-) stretching and thus confirm the purity of the surfactant. Alternatively, dialysis in HFE7100 and methanol mixture would be another way to purify our surfactant if small molecules other than Krytox are the concern. However, we did not try that as repeated washing of the deprotected surfactant yielded slightly yellowish to colorless product with sufficient purity.

We have improved the manuscript with a brief discussion in the revised manuscript on page 4.

3. As the authors mention, tri-block PEG-PFPE₂ surfactant will be a mixture of di-block and tri-block polymers due to incomplete reaction and uncertainty in molar ratio calculation. However, mixed surfactant system can often perform better in terms of droplet stability. Also, tri-block surfactant will have a denser polymer brush dissolved in oil phase, thus providing stronger barriers between droplets, all of which may explain the success of PEG-PFPE₂ surfactant. Why does a di-block polymer with a smaller head group and a smaller tail group perform better?

Response:

Till today, the tri-block copolymer PEG-PFPE₂ surfactant is the most successful one in droplet microfluidics. This surfactant uses PEG600 as the polar head group. Why a di-block version of it either with PEG300 or with PEG600 does not form a denser polymer brush and tolerate demanding reaction conditions such as PCR are still not well-explored. An excellent work by Baret et al. (NATURE COMMUNICATIONS, 2016 7, 10392; DOI: 10.1038/ncomms10392) showed that the tri-block based PEG-PFPE₂ surfactant assembles not only in a single micelle but also in a bilayer vesicle. This facilitates leakiness of the droplet contents. Thus, even though the generated droplets are stable, droplet integrity to provide single drop resolution data is often compromised.

It was a serendipity that we tried the dendritic oligo-glycerol to create fluorosurfactant even though our previous work (Lab Chip, 2016, 16, 65-69; DOI:10.1039/C5LC00823A) that employed linear polyglycerol head group could not stabilize droplets at high temperatures. Instead, the methylated linear polyglycerol worked out. We reason that the presence of hydroxy groups in the linear polyglycerol provides irregular orientation of the hydrogen bonding along with the possibility of forming a bilayer vesicle. This suggests that the geometry of the head group is crucial. In another work (Chem. Eur. J., 2016, 22, 5629-5636; doi.org/10.1002/chem.201504504) we showed that when dendritic tri-glycerol is used to create an amphiphile to form classical micelles, the hydroxy groups form inter-and intra-molecular hydrogen bonds and their self-assembly can be tuned to adopt to different architectures depending on the type of chiral centers present. We believe that the tendency to form spontaneous and optimum inter-and intra-molecular hydrogen bonds significantly influence the performance of the di-block fluorosurfactant even though the MW of the head group is three times less than PEG600 – the most successful head group reported so far in synthesizing a tri-block copolymer fluorosurfactant.

4. The authors mention, "We created droplets using these surfactants at roughly equimolar amounts, corresponding to 2% w/w H-dTG and 0.7% w/w L-dTG. Under these conditions, H-dTG droplets were more stable than L-dTG-stabilized droplets, suggesting that the H-dTG longer fluorinated tails can provide more stability than shorter tails (data not shown). Of note, droplets

made with 2% w/w L-dTG were more stable than those formed with the PEG-PFPE2 surfactant (Fig. 2c)." This reviewer thinks that comparing surfactant performance with samples containing equimolar surfactant concentration is a fairer comparison. It appears that 2% w/w final surfactant concentration is simply an experimental choice (a concentration that does not affect viscosity of oil phase too much) and can't serve as a criteria for comparing surfactant performance. The optimal surfactant concentration can be different for each type of surfactant. Based on estimated molecular weights (15 kD for PEG-PFPE2, 7.5 kD for di-block H, 4 kD for di-block M, 2.5 kD for di-block L), the molar concentration can vary as large as 6 fold (e.g., between PEG-PFPE2 and di-block L) for the same w/w percentage concentration. I suggest that the data on emulsion stability under PCR condition at equimolar concentration (corresponding to 5% w/w PEG-PFPE2 concentration) be included in the revised manuscript.

Response:

A detailed discussion is given below, and we have improved the manuscript with the discussion in revised manuscript on pages 6, 10, 14, and 15.

We did test equal molar concentration of 2% w/w H-dTG and 0.7% w/w L-dTG and mentioned their stability. This concentration corresponds to 4% w/w PEG-PFPE₂ at equal molar concentration, which is similar to the concentration suggested by the reviewer. As suggested, we added emulsion stability under PCR data of 4% PEG-PFPE₂ in supplement information (Supplementary Fig. 2-4) and revised corresponding paragraph on page 6. From our result, drop stability of 4% w/w PEG-PFPE₂ is similar to 2% w/w H-dTG and both are better than 0.7% w/w L-dTG.

We think it is hard to decide whether it is fairer to compare surfactant performance with equal molar concentration or equal w/w concentration. In our case, viscosity of the neat larger molecular weight surfactant either PEG-PFPE₂, H-dTG or M-dTG is higher than smaller molecular weight surfactant L-dTG. Thus, we should expect the highest molar concentration we can work with of L-dTG is higher than that of PEG-PFPE₂.

Another reason why we choose to use same w/w concentration for surfactant comparisons is cost, which is often a major part of microfluidics systems. In this case, no matter what the length of PFPE is, the cost of small, medium, and large PFPEs per gram is the same. From this perspective, it is appropriate to compare surfactant with same w/w concentration.

5. Have you tried Taqman-type PCR with your surfactant? The probe generates fluorescent dye molecules as a result of polymerase activity and therefore such an in-droplet reaction and fluorescence imaging before and after PCR would be a great demonstration of both droplet stability and inter-droplet dye transfer rate.

Response:

We decouple these experiments in our work which gives us results on droplet stability and cross-talk of dye separately. However, we agree with the reviewer that TaqMan-type PCR can be a powerful tool which provides results on both topics and makes the experiments more concise. We will try this out in our future experiments.

6. You mentioned that hydrogel droplets are generated at kHz rates (Supplementary Figure 5) and gels were transferred to aqueous solution. Making droplets of highly viscous polymer at a high speed is challenging, but this reviewer could not find any information on the device and flow rates used, nor the work-up procedure for recovering hydrogel particles. Please add more detailed information, which would be useful to the interested readers.

Response:

We added the work up procedure in SI on page 5 entitled ‘release of microgel particles’ and the information on the device and flow rates in the supplementary fig. 6.

7. Even though the choice of fluorescein and doxycycline makes sense, this reviewer disagrees with labeling them as water-soluble and DMSO-soluble. Fluorescein is soluble both in water and DMSO. The same is true for doxycycline. According to the literature (<https://www.ncbi.nlm.nih.gov/pubmed/34018> and <https://pubs.acs.org/doi/abs/10.1021/ie060055v>), doxycycline and its more well-known cousin tetracycline dissolve in water fairly well (up to ~40 mM). It is common practice to dissolve antibiotics or inducers in DMSO at high concentration and dilute to culture media, but such labeling might give a false impression that fluorescein and doxycycline are representative of hydrophilic and hydrophobic molecules.

Response:

We would like to thank the reviewer for providing the references. It is worth mentioning that the solubility of drug doxycycline in water, in PBS buffer (pH:7.4), or in cell culture medium could be different. According to the literature entitled ‘Solubility of Doxycycline in Aqueous Solution’ (Journal of Pharmaceuticals Sciences, 1979, 68, 188-194; doi.org/10.1002/jps.2600680218), the solubility of the protonated form of doxycycline is clearly influenced by both acidic medium and salt concentration. In our study, when the M-dTG surfactant is used to stabilize the droplets, the inter-droplet exchange of water-soluble dye is minimal even at a time scale of order several days. More broadly, even though the cell culture medium contains 10-15% FBS, a protein known to slow down the solute exchange rate, DOX retention ability of the M-dTG surfactant stabilized droplets is different than what we see when the water-soluble fluorescein dye is encapsulated. This suggests that DOX’s solubility in cell culture medium could be different than that in regular water. However, a detailed study is needed to draw a clear conclusion on this matter. With this in mind, we labelled the drug DOX.HCl as DMSO-soluble DOX. In fact, DOX is sold in different forms including DOX Hyclate (readily soluble in water) and DOX. H₂O (poorly soluble in water). No matter which derivative we use, the DOX induced GFP expression is the same under the cell culture conditions. Since labelling of DOX.HCl as ‘DMSO-soluble DOX’ might be misinterpreted, as suggested, we mentioned it in our revised manuscript as just ‘DOX’.

8. Fluorescein is a polar molecule with a phenol and a carboxy group and doxycycline has 5 hydroxy groups and one amide bond, which makes them efficient participants to hydrogen bonding network. This may limit the applicability of the surfactant to hydrogen bond-capable molecules. Fluorescence imaging data of inter-droplet transfer for more leaky dyes such as resorufin and Nile

red would be informative and better define the performance gains and applicability of the dendronized di-block surfactant.

Response:

We tried to work with Nile red to investigate the inter-droplet transfer. However, this dye precipitates when dissolved in PBS buffer and if a nanocarrier such as SDS is used to dissolve this hydrophobic dye in water, the inter-droplet transfer happens during droplet preparation (data not shown). This suggests that for extremely hydrophobic dye our dendronized surfactant alone cannot prevent the inter-droplet leakage. A better carrier such as PLGA-based nanoparticles to encapsulate Nile red would be more rational. However, we wanted to elucidate the performance of our dendronized surfactant without adding any additives that can solubilize the solute in aqueous phase better and reduce/prevent the inter-droplet transfer. Instead, we worked with resorufin dye that does not need any additive to get it solubilized in aqueous phase. We show that when M-dTG surfactant, a representative and the best performer among the oligo-glycerol based surfactants, is used, the inter-droplet transfer of this dye is minimal till 30 min. In contrast, almost 50% dye is exchanged when PEG-PFPE₂ surfactant is used. However, when M-dTG surfactant is used, the inter-droplet transfer of ~50% resorufin dye occurs after 330 min, indicating that the inter-droplet transfer of resorufin in M-dTG stabilized droplets is 11 times slower than PEG-PFPE₂ stabilized droplets (Supplementary Fig. 10). We revised the corresponding paragraph on page 10.

9. How sensitive is the GFP reporter system within droplets? 90% cell survival rate in droplet culture does not mean that their GFP protein synthesis is as efficient as the bulk cell culture. I could not find the data showing the sensitivity/efficiency of GFP induction when the doxycycline is present in the droplet. Preferably, the GFP-positive fraction of cells should be compared between bulk culture and droplet culture at the same doxycycline concentrations to ensure that the effect shown in Figure 5 (for the DOX diffusion column) is mainly due to prevention of inter-droplet transfer, rather than the result of direct impediment of GFP induction by droplet environment.

Response:

To find out the most sensitive reporter cell line we again screened the transfected cells. Hence, we worked with a new batch of the cell lines to redo the experiment reported in Figures 4a and 5 and added the updated data in Figures 4a and 5 on the revised manuscript. With this cell line we could achieve a maximum of 73.5% GFP⁺ cells after 48h when they were incubated with 500 nM DOX. However, the author of the XLone-GFP plasmid construct reported ~77% GFP⁺ cells with human pluripotent stem cell lines (hPSCs).

We compared the GFP-positive fraction of cells between bulk culture and droplet culture at the same DOX concentration. Indeed, droplet environment does not interfere the GFP expression of cells (compare Fig. 5 column 2 with column 3). However, we do see that the number of GFP⁺ cells in droplet is marginally less than in bulk culture. We rationalize this observation by noting that the effective concentration of cells in bulk culture is ~10 times less than in droplet culture, but the molar concentration of DOX is the same, that leads to a high drug to cell ratio in bulk culture compared to the droplet culture. Consequently, the number of GFP⁺ cells in bulk culture is slightly higher than in droplet culture.

Further, when cells are incubated with DOX in 4% (w/w) PEG-PFPE₂ surfactant stabilized droplets, we see the same trend in GFP expression level as in droplets stabilized with M-dTG surfactant (compare Fig. 5 column 3 with Supplementary Fig. 12 column 2). We have improved the manuscript with Fig. 5 and Supplementary Fig. 12 and made corresponding statements on pages 13-17.

10. In Figure 3b, why would the average intensity go down from Day 2 to Day 3 for PEG-PFPE₂ sample? Shouldn't the fluorescence intensity of PBS-only droplets go up until it saturates to the maximum value (theoretically, half of the intensity from the initial dye-containing droplets)? Due to high variability in fluorescence intensity in the droplet population, it may be challenging to distinguish PBS-only droplet and dye-containing droplet. What were the selection criteria for image analysis? Were there two distinct populations over the course of experiments?

Response:

To distinguish two droplet populations in a mixture, one straightforward approach would be to generate droplets of two different sizes. Instead, we prepared droplets of identical size using a parallel drop maker to avoid any potential influence of size dependent inter-droplet leakage from a smaller dye-containing droplet to a bigger no-dye-containing droplet and vice-versa. Hence, it is challenging to distinguish PBS-only droplet and dye-containing droplet when droplets are of identical size and all the droplets become nearly saturated to the maximum value which is half of the intensity from the initial dye-containing droplets with fluorophore. We used the 'line profile' tool of Leica software to quantify the green fluorescence intensity of the PBS-only-droplets. We selected relative dim droplets from the droplet mixture. It is worth mentioning that although the average fluorescence intensity for PEG-PFPE₂ sample goes down from Day 2 to Day 3, the error bar in Day 3 data is within the range of Day 2 data. However, imaging of the same droplets trapped in a microchannel from Day 1 to Day 3 would be much more precise.

Reviewer #2

(1) The authors stated that the length of fluorinated tails would alter the droplet stability (Figure 2, and in the paragraph stating that the H-dTG can provide more stability than L-dTG). However, the presented data is confusing. If the reviewer has gathered the information correctly, the surfactant molecules on the oil-drop interface matters, given that the H-dTG droplets were observed more stable than the L-dTG one at the equimolecular condition?

Response:

With equimolar concentrations of H-dTG and L-dTG surfactants we wanted to see i) the droplet stability and ii) decouple the influence of long vs short PFPE chain on shell strength from the influence of polar head group. Indeed, we see that the shortest PFPE chain does not provide enough strength to the droplet shell even if the molar concentration is the same as in H-dTG. We also observe that the viscosity of neat larger molecular weight surfactant (PEG-PFPE₂, H-dTG, M-dTG) is higher than smaller molecular weight surfactant L-dTG. This may partially explain why droplets stabilize with less than 1% (w/w) L-dTG surfactant are not robust. Instead, the higher is

the concentration of L-dTG, the better is the droplet stability. We made a brief corresponding statement in revised manuscript on page 6.

Yet, moving on to the description of Figure 3, the best performing group became M-dTG? If this was determined by the inter-droplet leakage, the data for H-dTG and L-dTG shall be presented as a comparison as well.

Response:

We presented the data for H-dTG and L-dTG in the Supplementary Fig. 8 because of the following reason. Prior studies showed that longer PFPE chain reduces the inter-droplet exchange. In our study, we found that the more is the number of hydroxy groups in the polar head group, the less is the inter-droplet transfer. Hence, we wanted to validate our claim by showing that number of hydroxy groups does play a key role in inter-droplet transfer even if the PFPE chain length is the same.

(2) The authors claimed that the improved thermal stability and minimized inter-droplet transfer of molecules (FITC and doxycycline) of the dTG droplets may be attributed to the inter- and intramolecular hydrogen bonds. Any direct evidence to prove it? For example, oxidize the hydroxyl groups to carboxyl group to reduce the strength of hydrogen bonding, or shield the inter- and intramolecular hydrogen bonding in H-dTG, MH-dTG and L-dTG by some small molecules.

Response:

To demonstrate the hydrogen bond donating activity of the hydroxyl groups in the M-dTG surfactant, a representative of the oligo-glycerol-based surfactants, we tested if high salt concentration in PBS medium can create ion-dipole interactions and, consequently, disrupt the inter- and intramolecular hydrogen bonding in M-dTG by the salt ions. Indeed, in presence of 5M NaCl we see inter-droplet leakage of water-soluble fluorescein dye after 24 h (Supplementary Fig. 9). The corresponding text is now revised on page 10.

(3) In line with the previous point, the length of fluorinated tails of the PEG600-based surfactant shall be clearly described. Is it H-PFPE as illustrated in Figure 2b?

Further, was the concentration of PEG-PFPE₂ also 2% w/w throughout the manuscript? The equimolar counterpart shall be contrasted with dTG-surfactants for a fair comparison.

The PEG-PFPE₂ based surfactant is commercially available by different vendors. Even the same vendor, for example ShpereFluidics, there are different kinds. Which commercially available surfactant has this study employed?

Response:

We purchased the tri-block copolymer fluorosurfactant PEG-PFPE₂ (EA surfactant) from RAN Biotechnologies. The EA surfactant has two perfluoropolyether tails (each having a MW ~6000 g mol⁻¹) coupled to a homo-bifunctional PEG600-amine head group.

We used 4% (w/w) PEG-PFPE₂ surfactant to show post-PCR droplet stability, inter-droplet transfer of resorufin dye and DOX. Even though this concentration does not represent the

equimolar concentration of all the oligo-glycerol based surfactants except H-dTG surfactant, we assume this is good enough to contrast with our new surfactants.

Further, the reason why we choose to use 2% w/w concentration for surfactant comparisons is cost, which is often a major part of microfluidics systems. In this case, no matter what the MW of PFPE is, the cost of small, medium, and large PFPEs per gram is the same. From this perspective, it is fairer to compare surfactants with the same w/w concentration.

Thanks again and we revised the manuscript accordingly on pages 2, 6, 10, 14-17.

(4) The droplet stability may be intuitively understood by the bright field images as shown in Figure 2 or Supplementary Figure 3. But it may provide a quantitative understanding by the variation of droplet diameters as shown in Figure 4d. Also, how many droplets that the study have investigated for each condition?

(5) Proper statistical analysis shall be conducted to validate the difference as claimed.

Response for 4 and 5:

We performed quantitative analysis of droplet size distribution and included the data in supplementary fig. 4. We used ~100 droplets for Pre-PCR size distribution analysis and ~200-400 droplets for Post-PCR size distribution analysis.

• Inter-droplet DOX transfer:

(1) If the minimized inter-droplet transfer of dTG-based surfactant is due to the hydroxyl groups, the data presented with water-soluble fluorescein makes sense. But it may not necessarily true for the DMSO soluble or hydrophobic drugs such as DOX. There is only data of M-dTG droplets but not the PEG-PFPE₂ ones. The comparison of the two types of surfactant for their inter-droplet DOX transfer is considered necessary.

Response:

The inter-droplet exchange rate of DOX seems to be quite similar to the exchange rate of resorufin dye. As suggested, we added the data of PEG-PFPE₂ surfactant (Supplementary Fig. 12). We would like to mention that although our M-dTG surfactant cannot significantly prevent inter-droplet transfer of DOX like it does in case of water-soluble fluorescein dye, it does slow down the transfer rate which is better than PEG-PFPE₂ surfactant. When M-dTG surfactant is used, 37.1% of the droplet-encapsulated cells become GFP⁺ when incubated with 1 μ M DOX-droplets for one day. In contrast, 44.5% of the droplet-encapsulated cells are GFP⁺ when incubated with 1 μ M DOX-droplets stabilized PEG-PFPE₂ surfactants (Fig. 5 and Supplementary Fig. 9).

We revised the corresponding statements and paragraphs on pages 13-17.

(2) The authors claimed that HEK 293 cell survival was above 90% in single emulsions for 3 days culture, which is quite surprising. It is recommended to test the cell viability with dual fluorescence staining such as Calcein-AM and PI, as Trypan Blue staining result is known to be not as accurate and may be affected by the staining protocols.

Response:

In our revised work we tested the cell viability with dual fluorescence staining kit (Invitrogen). We did not see a significant difference in HEK293 cell viability (Fig. 4e). It is known that cell lines such as K562, HEK293 are quite robust. This might explain why their survival in droplet culture is so high. In our previous work we also observed that the cell viability is quite good when linear triglycerol-based fluorosurfactant was used and cells were cultured on HFE7500 oil (Lab Chip, 2016, 16, 65-69; DOI:10.1039/C5LC00823A). We think that, in addition to compartmentalization, the HFE7500 oil does favor the cell survival by i) providing more oxygen to cells as they carry a lot of dissolved oxygen and ii) dissociating acidic ions into the oil phase when the cell culture medium becomes more acidic.

Further, since we added the commercial surfactant data in the revised manuscript, we are now replacing the previous cell survival data with the cell survival data from standard culture (positive control), M-dTG surfactant, and PEG-PFPE₂ surfactant on page 14 (Fig. 4e).

• Has the interfacial tension with the dTG-based surfactant been characterized, since interfacial tension would also interfere the droplet size and stability?

Response:

We have not measured the interfacial tension in our study. We do agree that the interfacial tension might interfere the droplet size and stability. However, we think interfacial tension is not the only parameter that can significantly dictate the fate of the droplet. We rationalize this hypothesis by noting that prior studies suggest geometry of the polar head group plays a crucial role to determine the droplet fate. In our previous work (Lab Chip, 2016, 16, 65-69; DOI:10.1039/C5LC00823A), we investigated linear polyglycerol and its alkylated version to prepare fluorosurfactant. We found that even though the polar group had ~13 hydroxy groups, it could not make stable droplets. Instead, the alkylated derivative worked out. This observation predicts that unless a suitable geometry is chosen, preparation of stable water-in-oil emulsion would be extremely challenging. We will address the interfacial tension of our surfactant in our future work.

We again thank both editor and reviewers for the very fast, professional, and supportive discussions of our manuscript and hope that the manuscript can be accepted in the present form.

With kind regards,

David A. Weitz & Rainer Haag

REVIEWERS' COMMENTS:

Reviewer #1 (Remarks to the Author):

This reviewer is satisfied with the revised manuscript and thinks that this manuscript is now suitable for publication.

Samuel Kim

Reviewer #2 (Remarks to the Author):

Thanks the authors' effort. The revision has well addressed the reviewer's comments. Only one minor comment to the authors:

The added paragraph on page 4 of the manuscript: "THF" shall be spelled out.